# Landscape of TB Infection and Prevention among People Living with HIV

**DOI:** 10.3390/pathogens11121552

**Published:** 2022-12-16

**Authors:** Anca Vasiliu, Rebecca Abelman, Yousra Kherabi, Antonia Morita Iswari Saktiawati, Alexander Kay

**Affiliations:** 1Global TB Program, Department of Pediatrics, Baylor College of Medicine, Houston, TX 77030, USA; 2Division of HIV, Infectious Diseases, and Global Medicine, Department of Medicine, University of California San Francisco, San Francisco, CA 94110, USA; 3Department of Infectious Diseases, Hôpital Bichat-Claude Bernard, Assistance Publique Hôpitaux de Paris, 75018 Paris, France; 4Centre d’Immunologie et des Maladies Infectieuses, (Cimi-Paris), INSERM U1135, Sorbonne Université, Équipe 2, 75013 Paris, France; 5Department of Internal Medicine and Center for Tropical Medicine, Faculty of Medicine, Public Health and Nursing, University Gadjah Mada, Yogyakarta 55281, Indonesia; 6Baylor College of Medicine Children’s Foundation-Eswatini, Mbabane H100, Eswatini

**Keywords:** tuberculosis prevention, PLHIV, TPT regimens, symptom screening, TB infection

## Abstract

Tuberculosis (TB) is one of the leading causes of mortality in people living with HIV (PLHIV) and contributes to up to a third of deaths in this population. The World Health Organization guidelines aim to target early detection and treatment of TB among PLHIV, particularly in high-prevalence and low-resource settings. Prevention plays a key role in the fight against TB among PLHIV. This review explores TB screening tools available for PLHIV, including symptom-based screening, chest radiography, tuberculin skin tests, interferon gamma release assays, and serum biomarkers. We then review TB Preventive Treatment (TPT), shown to reduce the progression to active TB and mortality among PLHIV, and available TPT regimens. Last, we highlight policy-practice gaps and barriers to implementation as well as ongoing research needs to lower the burden of TB and HIV coinfection through preventive activities, innovative diagnostic tests, and cost-effectiveness studies.

## 1. Introduction

An estimated 9.9 million people suffered from tuberculosis (TB) in 2020 [1], and a quarter of the people worldwide have been infected with latent TB [2]. People living with HIV (PLHIV) represented 8% of TB cases and contributed to approximately 14% of TB deaths in 2020 [1]. Moreover, TB represents the leading infectious cause of death among PLHIV [3,4]. TB mortality in PLHIV is as high as 30% worldwide [4]. Countries in the World Health Organization (WHO) African Region have the highest TB-HIV coinfection rate. In sub-Saharan Africa, PLHIV with TB have a 40% higher mortality than those without TB [1,5].

In addition to higher mortality, PLHIV are simultaneously more likely to develop TB and are less likely to be diagnosed. PLHIV are 19 times more likely to develop TB disease than their seronegative counterparts [6], and global estimates indicate a higher case detection gap among coinfected individuals [1]. To address these gaps, current WHO guidelines aim to target early detection and treatment of TB among PLHIV, particularly in high-prevalence and low-resource settings. BCG vaccination is not recommended in infants living with HIV due to the risk of BCG disease unless they are immunologically stable [7]. The WHO recommends using TB preventive treatment (TPT) as primary prevention in newly diagnosed PLHIV and secondary prevention in PLHIV who have been exposed to TB, irrespective of their antiretroviral treatment (ART) status [8]. To be eligible for TPT, active TB disease should be excluded first [8]. PLHIV may also receive TPT after completing TB treatment [8]. Adults and adolescents may take TPT regardless of their immune and pregnancy status [8]. There are many options available for TPT. In its 2020 guidelines, the WHO recommended the use of three months of daily isoniazid with rifampicin (3 HR), three months of weekly isoniazid plus rifapentine (3 HP), or along with six to nine months of isoniazid (6–9 H) as the preferred TPT regimens regardless of HIV status [8]. One month of daily isoniazid plus rifapentine (1 HP) and four months of daily rifampicin (4R) were considered acceptable alternative regimens. 

Despite several options for TPT, gaps remain in diagnosis and policy uptake. Between 2005 and the end of 2020, 13 million PLHIV were initiated on TPT, equivalent to about one-third of the 37.7 million people estimated to be living with HIV in 2020 [1]. Coverage varies widely among countries, with a median coverage of 54% (interquartile range [IQR], 41–79%) among eligible people newly started on ART [1]. 

## 2. Screening for TB

### 2.1. Symptom-Based Screening

The foundation of screening for tuberculosis among PLHIV in high-incidence settings is the symptom-based screen. Current WHO guidelines recommend that adults and adolescents with HIV receive symptom-based screening for TB at each visit to a health facility [6]. The WHO four-symptom screen evaluates the presence of current cough, fever, weight loss, or night sweats. This evaluation is preferred as a first test given its negative predictive value, allowing for the exclusion of active TB prior to the initiation of TPT. Prior meta-analyses determined that the absence of fever, cough of any duration, night sweats, or weight loss among PLHIV had a negative predictive value of 97.7% (95%CI: 76.1–94.4%) [9]. More recently, the WHO performed a meta-analysis including 23 studies that found the pooled sensitivity among all PLHIV was 83% (95%CI: 74–89%) and the specificity was 38% (95%CI: 25–53%) [6,10]. Notably, performance was lowest among outpatients on ART and pregnant women. Those with a positive symptom-based screen are subsequently referred for further evaluation of TB disease. 

For children with HIV, screening for tuberculosis is complicated by less specific symptoms in children and lower sensitivity of existing diagnostic tests [11]. However, symptom screening remains a key component of TB screening in the WHO guidelines for children with HIV, with a positive symptom screen defined as any fever, cough, poor weight gain, or recent TB contact [12]. While there are fewer data on children with HIV on the performance of symptom screening, evidence suggests that the symptom screen has high specificity but low sensitivity [11,13,14,15].

### 2.2. Chest Radiography

Chest radiography (CXR) is a useful tool for ruling out active tuberculosis among PLHIV. CXR was a new addition to the WHO guidelines in 2021, largely because of its negative predictive value. In a meta-analysis of eight cross-sectional studies, any abnormality on CXR or the four-symptom screen used in parallel yielded a sensitivity of 93% (95%CI: 88% to 96%) and specificity of 20% (95%CI: 10% to 38%) as compared to a culture reference standard [6,16,17,18,19,20,21,22,23]. In a prevalence study by Modi et al. in Western Kenya, the addition of CXR increased symptom-based screening sensitivity from 74.1% (95%CI: 64.1% to 82.2%) to 90.9% (95%CI: 86.4% to 93.9%) and had a negative predictive value of 96.1% (95%CI: 94.4% to 97.3%) [19]. These discrepant results are likely partially attributable to differences in TB prevalence between studies. For example, the 2018 WHO grade assessment determined that at a 5% TB prevalence among PLHIV, CXR increased the negative predictive value by only 1%, whereas at a TB prevalence of 20%, the negative predictive value increased by nearly 4% (90.4% vs. 94.3%) [10]. These results suggest that, where available, CXR likely serves as a useful tool for diagnosing TB among PLHIV but is of the highest utility among outpatients in high-prevalence areas. There is limited data on using CXR among hospitalized PLHIV, but the combined strategy of the symptom screen and CXR has been found to have low specificity [23]. Nevertheless, access to CXR should not be a barrier to initiating TPT, especially for children under 5 living with HIV, as shown by a meta-analysis evaluating CXR compared to symptom screening in excluding TB, which found that symptom screening alone can be used in excluding TB in children under 5 years with and without HIV in high-burden settings [24]. In a Cochrane Review, a meta-analysis on the performance of abnormal CXR among children with and without HIV with a close TB contact found a pooled sensitivity of 87% and a pooled specificity of 99% [11]. Notably, this included studies conducted in low-incidence settings.

### 2.3. Biomarkers—CRP

Several studies have evaluated the utility of C-reactive protein (CRP) in screening for TB disease among PLHIV. A biomarker of general inflammation, cutoff values of >5 mg/L and >10 mg/L have been investigated as a tool for TB diagnosis given the ability of CRP to be measured using point-of-care tests and by capillary blood collected by a finger prick [17,25,26]. Among PLHIV it has utility, particularly among outpatients not on ART. A grade analysis by the WHO found that when compared with culture as a reference standard, CRP with a cutoff of 5 mg/L had a sensitivity of 90% (95%CI: 78–96%) and a specificity of 50% (95%CI: 29–71%) [6]. Compared to the symptom-based screen, the cutoff value of >5 mg/L was felt to have similar accuracy and, in some sub-populations, had a higher specificity than symptom-based screening alone [6,23]. It has a poor specificity (12%, 95%CI: 9–17%) among inpatients with HIV, likely due to competing comorbidities that would also raise CRP levels [6,23]. Where available, this should be considered as an additional test to enrich screening for TB among outpatient PLHIV.

### 2.4. TST, IGRAs, Xpert and LAM

The use of tuberculin skin testing and interferon-gamma release assays are of lower utility among PLHIV in high TB incidence areas. Tuberculin skin testing (TST) has many operational challenges in low-resource settings; patients are required to return 48–72 h after placement, and in patients with advanced HIV, anergy may result in false negative results [27]. While interferon-gamma release assays (IGRA) have better performance than TST, higher rates of indeterminate test results were found with Quantiferon Gold in PLHIV when compared to their seronegative counterparts [28,29,30]. Low CD4 counts contribute to T-cell anergy that may also influence IGRA results, although to a lesser degree than TST [28,31,32,33]. To date, there is little data evaluating the performance of IGRA for the diagnosis of latent TB among PLHIV in high-incidence settings [10]. For these reasons, IGRA and TST results are not required prior to TPT initiation in the WHO guidelines. Instead, other testing methods, such as the Xpert/RIF or LAM assay, are recommended by the WHO for high-risk patients, such as hospitalized PLHIV [10]. In low TB incidence areas, IGRA tests are still recommended for TB screening and TPT initiation [34,35]. 

Other methods, such as urine lipoarabinomannan (LAM) testing, have shown promising results for determining TB status among PLHIV, but further research is needed [21,36]. Urine LAM testing is recommended by the WHO for PLHIV in outpatients with CD4 < 100 and inpatients with CD4 < 200, even in the absence of TB symptoms [23].

## 3. TPT Regimens

TPT has been shown to reduce the progression to active TB and mortality among PLHIV, regardless of the concurrent receipt of ART [37]. A systematic review found that TPT reduced the overall risk for TB by 33% [38], with a reduction increase of 64% among those with a positive TST [38]. In addition, a randomized controlled trial showed that TPT reduced the risk of severe HIV-related illness and TB mortality, even in PLHIV with high CD4+ cell counts, with a protective effect that lasted more than 5 years [38,39]. TPT should be used only after the exclusion of active TB disease. 

Several TPT regimens are available for PLHIV. Some are based on isoniazid only (H) with 6 to 9, 12, and 36 months of treatment (6–9 H, 12 H, and 36 H). Others combine isoniazid with rifamycin, i.e., three months of isoniazid plus rifampicin (R) and one or three months of isoniazid plus rifapentine (P). To date, there are no direct comparisons between all these TPT regimens published in the literature. We summarized the literature on the regimens with and without ART and completion rates in Table 1.

Literature on TPT in children living with HIV (CLHIV) is scarce (Table 1). Current WHO and CDC guidelines both recommend isoniazid-based regimens and 3 HR for TPT in all CLHIV [40,41].

**Table 1 pathogens-11-01552-t001:** Main characteristics of studies assessing tuberculosis preventive therapy effectiveness in people living with HIV in Low-or-Middle-Income Countries.

Reference	Country or Area	Number of Participants Included	Adults/ Children	TPT Regimen (Months/Drug)	Number of Patients on ART	Type of ART	Primary Outcome	Comple- tion Rate	Main Results	Conclusion
Hawken et al., 1997 [42]	Kenya	684	Adults & Children ≥ 14 years-old	6 H Placebo	NR	NR	Death OR Development of TB	NR $	Development of TB adjusted rate ratio 6 H: 0.92 [0.49–1.71] Mortality adjusted rate ratio 6 H: 1.18 [0.79–1.75]	No statistically significant protective effect of 6 H.
Whalen et al., 1997 [43]	Uganda	2736	Adults	6 H 3 HR 3 HRZ Placebo	0	NA	Development of TB	92% in 6 H 93% in 3 HR 90% in placebo	Relative risk for TB development (compared to placebo) 6 H: 0.33 [0.14–0.77] 3 HR: 0.40 [0.18–0.86] 3 HRZ: 0.51 [0.24–1.08]	6 H, 3 HR, and 3 HRZ reduce the risk of developing TB compared to placebo. Adverse events more frequent in 3 HR and 3 HRZ regimens.
Gordin et al., 2000 [44]	USA, Mexico, Brazil	1583	Adults & Children ≥ 13 years-old	12 H 2 RZ	574 (36.3)	NR	Development of culture-confirmed TB	69% in 12 H 80% in RZ	Risk ratio for TB development 12 H: 0.72 [0.40–1.31]	2 RZ is similar in safety and efficacy to 12 H.
Fitzgerald et al., 2001 [45]	Haiti	237	Adults	12 H Placebo	0	NA	Development of TB AND/OR Death AND/OR AIDS	NR	Relative risk for TB development 12 H: 1.26 [0.36–4.37] Relative risk for death 12 H: 1.12 [0.67–1.87] Relative risk for AIDS 12 H: 1.05 [0.55–2.03]	No statistically significant protective effect of 12 H.
Zar et al., 2007 [46]	South Africa	263	Children (≥8 weeks)	24 H (daily and t.i.w) Placebo	81 (30.8%)	NR	Development of TB AND/OR Death Adverse events	NR	Development of TB 24 H: Hazard ratio 0.28, [0.10–0.78] Mortality 24 H: Hazard ratio 0.46 [0.22–0.95]	24 H reduces mortality and TB development as compared to placebo.
Madhi et al., 2011 [47]	South Africa Botswana	548	Children between 91st and 120th day of life	24 H Placebo	542 (98.2)	NR	Development of TB AND/OR Death	NR	Combined incidence of tuberculosis infection, tuberculosis disease, or death 24 H: 19% Placebo: 19.3% (*p* = 0.93)	No statistically significant protective effect of 24 H.
Martinson et al., 2011 [48]	South Africa	1148	Adults	3 HR 3 HP 72 H 6 H (control)	215 (18.7)	NR	TB-free survival	83.8% in 6 H 95% in 3 HR 96% in 3 HP 43% in 72 H	Incidence rates of active tuberculosis or death were (in person-years): 3 HR: 2.9/100 3 HP: 3.1/100 72 H: 2.7/100 6 H: 3.6/100 (*p* > 0.05 for all comparisons)	All 4 regimens were effective. Neither 3 HP, 3 HR, nor 72 H were superior to 6 H.
Samandari et al., 2011 [49]	Botswana	1995	Adults	6 H 36 H	946 (47.4)	NNRTI-based ART *	Development of TB	NR	Development of TB (per year) 6 H: 1.26% 36 H: 0.72% Hazard ratio 0.57 [0.33–0.99]	36 H was more effective than 6 H.
Rangaka et al., 2014 [50]	South Africa	1369	Adults	12 H Placebo	1369 [100]	40 % D4T/3TC/EFV 33% D4T/3TC/NVP 18% AZT/3TC/NVP 9% Other	Development of TB	83% in 12 H 82% in placebo	Development of TB 12 H: Hazard ratio 0.63 [0.41–0.94]	Isoniazid TPT should be recommended to all patients receiving antiretroviral therapy irrespective of tuberculin skin test or interferon gamma release assay status.
Temprano study group 2015 [39]	Ivory Coast	2056	Adults	6 H No treatment	1630 (79.3)	55% TDF/FTC/EFV 18% TDF/FTC/LPV 6% Other 21% None	Death AND/OR AIDS AND/OR Cancer AND/OR Invasive bacterial infection	94% in 6 H	Primary composite outcome 6 H: adjusted hazard ratio 0.65, [0.48–0.88]	Immediate ART and 6 H led to lower rates of severe illness than did deferred ART and no TPT.
Badje (Temprano follow-up- 2017) [38]	-	-	-	-	1831 (89.1)	70% TDF/FTC/EFV 22% TDF/FTC/LPV 8% Other	All-cause mortality	-	6 H: adjusted hazard ratio 0.63 [0.41–0.97] after adjusting for the ART strategy (early vs. deferred)	6 H has a durable protective effect in reducing mortality in HIV-infected people
Sterling 2016 [51]	USA, Spain, Canada, Hong Kong, Brazil, Peru	399	Adults + Children ≥ 2 years-old	9 H 3 HP	125 (31.3)	Exclusion of regimens including PI NNRTI	Development of TB Completion Adverse reactions	64% in 9 H 89% in 3 HP	Cumulative rates of tuberculosis incidence: 9 H: 3.50% 3 HP: 1.01% (rate difference: −2.49%; upper bound of the 95% confidence interval of the difference: 0.60%). Drug discontinuation due to an adverse drug reaction 9 H: 4% 3 HP: 3% (*p* = 0.79)	3 HP was as effective and safe as 9 H, and better tolerated
Swindells 2019 [52]	10 countries in America, Asia, and Africa	3000	Adults & Children ≥ 13 years-old	9 H 1 HP	2891 (96.3)	NR	Development of TB AND/OR Death	90% in 9 H 97% in 1 HP	Development of TB and/or death (in person-years): : 1 HP: 0.65/100 9 H: 0.67/100 Rate difference, −0.02/100; upper limit of the confidence interval, 0.30	1 HP was non-inferior to 9 H in preventing TB.

$ «Tablet counts during the 6-month treatment period were available for 554 subjects, of whom 406 (73%) had an excess of seven or fewer tablets during the whole 6-month treatment period». * «Of 946 participants starting antiretroviral therapy, 415 (44%) received zidovudine, lamivudine, and nevirapine; 406 (43%) received zidovudine, lamivudine, and efavirenz; 52 (6%) received stavudine, lamivudine, and nevirapine or efavirenz; and 31 (3%) received tenofovir, emtricitabine or lamivudine, and nevirapine or efavirenz». Abbreviations. 3TC, lamivudine, ART, antiretroviral therapy; AZT, azidothymidine or zidovudine; D4T, stavudine; EFV, efavirenz; FTC, emtricitabine; H, isoniazid; LVP, lopinavir; NA, not applicable; NR, not reported; NVP, nevirapine; P, rifapentine; R, rifampin; RCT, randomized controlled trial; TB, tuberculosis; TDF, tenofovir disoproxil fumarate; t.i.w., three times a week; TPT, tuberculosis preventive therapy; Z, pyrazinamide.

### 3.1. Adherence and Completion

Despite studies supporting its efficacy, TPT uptake among PLHIV worldwide remains low. Between 2018 and 2020, out of the estimated 30 million eligible patients, less than 9 million PLHIV received TPT [53]. Several challenges have likely contributed to the low global uptake of TPT, such as adherence to long-duration regimens, drug–drug interactions with ART, and fear of the emergence of drug resistance [43,44,45].

Isoniazid-based TPT has been particularly challenging due to its long treatment course of 6 to 9 months resulting in low acceptance, adherence, and completion rates. While in controlled settings, such as clinical trials, good adherence was shown, in programmatic settings completion rates were only 50% to 70% of patients [54,55,56,57,58].

A high daily pill count may also be an obstacle, as multidrug therapies are prescribed for both TB and HIV, leading to a potential pill burden that may cause nonadherence to both these two therapies. Physicians taking care of PLHIV should give their patients the time to make informed decisions about what TPT regimen is right for them. In a cohort study including 586 patients in the UK, the odds of TPT completion increased 2.3-fold when subjects were offered a choice between two treatment regimens [59]. In a study led in Uganda, 252 PLVIH were randomized to a scenario of providing TPT by either directly observed therapy (DOT) or an informed choice between DOT and self-administered therapy. Patients assigned to DOT identified more barriers to completing TPT than those given a decision about their treatment [60]. 

With the advent of shorter and better-tolerated rifamycin-containing regimens, completion rates have increased [50,51]. For example, shortening treatment to 3 months with supervised administration of rifapentine and isoniazid increased completion rates to 96% in research settings and to 82% in practice [48,57,61]. In a 2019 randomized controlled trial in PLHIV comparing one month of daily rifapentine and isoniazid (1 HP) to nine months of isoniazid alone (9 H), Swindells et al. found not only was 1 HP noninferior to 9 H in TB prevention, but completion rates were higher in the 1 HP group (97% vs. 90%) [52]. Similarly, in a recent meta-analysis indirectly comparing TPT regimens in PLHIV, shorter rifamycin-containing regimens (3 HR, 3 HP, and 1 HP) seemed to have better completion rates than longer isoniazid-based regimens (6 H, 12 H, and 36 H) [62].

### 3.2. Drug-Drug Interactions

ART can prevent progression to TB disease by up to 65%; however, PLHIV remain at a high risk of developing active TB even with high CD4+ cell counts and provision of ART [63,64]. TPT and ART also have additive efficacy in preventing TB; therefore, the WHO recommends that they be given concurrently [39].

While shorter rifamycin-based TPT regimens are effective in PLHIV, the use of these regimens can be challenging due to the risk of drug–drug interactions with ART, as rifamycin is a potent inducer of cytochrome P450 [65]. For example, rifampicin can reduce protease inhibitors (PI) plasma concentrations up to 80–95% [66]. Several studies have shown that efavirenz is less prone to interaction with rifampicin compared to other ART; however, efavirenz is no longer considered as first-line therapy due to its side effect profile and concern for HIV drug resistance [58,59,60,67,68,69]. Dolutegravir-based regimens have emerged as first-line therapy, but dolutegravir requires twice daily dosing if used with rifampicin due to drug-drug interactions [70]. However, a phase I/II trial in South Africa showed that 12 doses of once-weekly rifapentine-isoniazid (3 HP) could be administered for TPT in PLHIV taking dolutegravir-based antiretroviral therapy without dose adjustments [71]. Data in children are still unavailable regarding the impact of the interaction between daily rifampicin or rifapentine, or weekly rifapentine on the efficacy and pharmacokinetics of dolutegravir.

While isoniazid is a potent inhibitor of cytochrome P450 enzymes, any potential inhibitory effect is most always counterbalanced by the stronger inductive effect of concomitantly administered rifampicin [72,73]. 

### 3.3. Side Effects and Concerns of Drug Resistance

Tolerance and side effects remain a major challenge to therapy completion among PLHIV. Both rifamycins and isoniazid have significant side effect profiles. Moreover, toxicity profiles of TPT and ART may overlap, and it is often difficult to define the drug involved in an adverse reaction. A typical example of such challenging situations is hepatotoxicity which can be due to ART, isoniazid, rifampicin, or their combination [65]. In a recent meta-analysis, rifamycin-containing regimens had a statistically significant lower occurrence of severe hepatotoxicity compared to isoniazid regimens [62]. Other side effects are cutaneous hypersensitivity, which can be caused by both isoniazid and rifampicin, and malaise associated with rifampicin use.

While the development of drug-resistant tuberculosis (DR-TB) is a concern among healthcare providers, there has been no data to suggest that TPT drives the emergence of drug-resistant tuberculosis, even in high-prevalence countries for DR-TB [48,74]. 

TPT can halt the progression to active TB for many years [75], but re-infection of TB after completing TPT can occur [76]. While TPT regimens can be changed following an adverse event or re-started after a brief treatment interruption, there is little data on the duration of administration of a new regimen after switching and how to manage interruptions of TPT (i.e., how long treatment should be prolonged for any specific number of missing doses without compromising the treatment efficacy) [9].

## 4. Policy-Practice Gap and Barriers to Implementation

Health professionals are often reluctant to prescribe TPT for PLHIV, even though guidelines in high-burden settings have adopted WHO recommendations for TPT initiation after a negative symptom screen among PLHIV, without the need for tests of infection [77]. Some reasons for their hesitancy come from the lack of confidence in excluding TB by symptom screening alone and the fear of promoting drug resistance [78].

The perceived fear of drug resistance has been repeatedly proven to be unfounded for isoniazid regimens [79]; nevertheless, it has been cited by healthcare workers numerous times as a source of hesitation for prescribing TPT [80]. 

The lack of integration between TB and HIV services serves as another barrier. HIV-specific programs promoting ART are moving towards differentiated service delivery, leaving TB services behind; TB services continue to be offered almost exclusively in a clinic-based model of care. There is an opportunity to align TB screening and TPT dispensation with ART services, but many TB programs have yet to seize that opportunity [81].

Health system shortcomings like drug stock-outs, reporting gaps, and lack of training for healthcare workers are widening the gap between policy and practice, making TPT less accessible to the key populations in need of TB prevention. Encouraging results can be obtained once TB prevention is prioritized, as shown by the United Nations High-Level Meeting successfully meeting their goal of 6 million PLHIV initiating TPT [1]. This remarkable achievement has been obtained with clear and strong WHO recommendations over many years, and donor support focused on TB prevention.

### Research Needs

Innovative diagnostic tests exploring the whole spectrum of TB, from latent infection to incipient, subclinical, and active disease, should be developed and made available for PLHIV. Shorter and long-acting regimens are needed to increase adherence and lower the pill burden for PLHIV. In addition, these regimens should come in a child-friendly formulation and be easy to administer by healthcare workers or caregivers, even to the youngest children. Children under 2 years should be included in randomized clinical trials evaluating TPT regimens to allow the formulation of robust evidence in this key population. Additional research is needed on 1 HP in children living with HIV who are under 13 years of age and on the safety of TPT regimens for pregnant women [52,62].

Drug-drug interactions with rifamycins should be explored, and TPT regimens adapted to the specific needs of PLHIV. In children, the interaction between TPT with 3 HP and dolutegravir-based ART is currently being explored by the DolPHIN kids study. Furthermore, TPT regimens for PLHIV exposed to drug-resistant TB should be developed, as coinfection with HIV and MDR/RR TB is associated with poor outcomes and increased mortality [82]. The cost-effectiveness of TPT for PLHIV has been proven numerous times in various settings [83]. Still, more research is needed on the cost-effectiveness of delivery strategies and the implementation of differentiated models of care. 

## 5. Conclusions

Encouraging progress has been made in TB prevention for PLHIV. Nevertheless, implementation gaps remain in diagnosis, treatment, access, and service delivery for this key population. Innovative research on TB infection and TB prevention for PLHIV is highly needed. To continue bridging the gap between policy and practice to make TPT accessible to all at-risk PLHIV, prevention should remain a research, policy, and donor priority.

## Data Availability

Not applicable.

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
