# Peer review of "Landscape of TB Infection and Prevention among People Living with HIV"

_pathogens, 2022, doi:10.3390/pathogens11121552_

Round 1

Reviewer 1 Report

This is a well written commentary about the status quo of TPT on PLHIV. The literature survey is exhaustive and the conclusions made are apt. I haven some minor comments. 

Please comment on the BCG vaccination status and its effect on Children with HIV? Is there a difference in TPT for these?

Line  119: Why does CRP perform poorly in inpatients with HIV? Please mention values if available.

What  could be the permissible gap in between two TPTs in a highly vulnerable population? 

I second the authors that latent TB diagnosis is a huge challenge and will influence the outcome of TPT.  Cost effective diagnostics and shorter and effective drug regimens are need of the hour.

Author Response

This is a well written commentary about the status quo of TPT on PLHIV. The literature survey is exhaustive and the conclusions made are apt. I have some minor comments.

We thank the reviewer for their thoughtful comments that improve our manuscript

Please comment on the BCG vaccination status and its effect on Children with HIV? Is there a difference in TPT for these?

Children living with HIV are at higher risk of BCG disease upon vaccination, but treatment with ART reduces this risk. We have added this information in the introduction

The risk of BCG disease is lower than the risk of not being vaccinated and getting severe/disseminated TB disease.

There is no difference in the choice of TPT according to BCG vaccination, but the vaccination should be proposed after TPT completion in CLHIV, as per the WHO recommendations.

Line 119: Why does CRP perform poorly in inpatients with HIV? Please mention values if available.

CRP has a low specificity in inpatients with HIV due to comorbidities that could also raise the CRP level. We added this information in the text.

What could be the permissible gap in between two TPTs in a highly vulnerable population?

This is an excellent question that we pondered upon as well, but there is no evidence in the literature at the moment, and no guidelines.

I second the authors that latent TB diagnosis is a huge challenge and will influence the outcome of TPT.  Cost effective diagnostics and shorter and effective drug regimens are need of the hour.

We thank the reviewer for his/her comment, and we agree that cost effective diagnostics as well as shorter and effective drug regiments are urgently needed.

Reviewer 2 Report

The authors did a nice work on summarizing the current studies and data of TB prevalence, TB test tools, TB prevention strategies, and challenges, especially for HIV infected people with ART. The authors cited sufficient references including the latest ones and highlighted the importance of developing innovative diagnostic tests and preventive ways and of bridging the gaps between the policy and practice. 

Minor comments:

Page 4 Table 1 is messy and needs adjustment.

Page 1 line36: delete the “in” of “Region in Have”

Page 2 line89: et al. 

Line 93: “grade” is not necessarily capitalized

Page 11 line278:  “dolphin” is not necessarily capitalized

Author Response

The authors did a nice work on summarizing the current studies and data of TB prevalence, TB test tools, TB prevention strategies, and challenges, especially for HIV infected people with ART. The authors cited sufficient references including the latest ones and highlighted the importance of developing innovative diagnostic tests and preventive ways and of bridging the gaps between the policy and practice.

We thank the reviewer for their nice and thoughtful comments

Minor comments:

Page 4 Table 1 is messy and needs adjustment.

We thank the reviewer for pointing this out, we will work with the journal to make Table 1 more reader-friendly

Page 1 line36: delete the “in” of “Region in Have”

We have deleted “in”

Page 2 line89: et al.

We added “.” for et al.

Line 93: “grade” is not necessarily capitalized

We have written grade in lower caps

Page 11 line 278:  “dolphin” is not necessarily capitalized

As this is the name of a trial and is an acronym for the study entitled “Safety and pharmacokinetics of dolutegravir in pregnant mothers with HIV infection and their neonates”, we will keep the capitalization, but write it “DolPHIN-1”, as the investigators proposed.

Reviewer 3 Report

Minor:

Line 32 – Perhaps specify latent infection vs active infection?

Lines 37-38 – “in sub-Saharan Africa” is redundant given context of sentence.

Line 40 – Sentence missing subject. Who is 19 more likely to develop TB?

Line 50 – “options for TPT” may read better as “options available for TPT”

Lines 87-88 and Lines 115-16 – Formatting of sensitivity and specificity inconsistent with rest of review. Is this how it was reported in the reference?

Line 91 – “and a negative predictive value” may read better as “and had a negative…”

Line 96 – Given context of sentence, “94.3% versus 90.4%” may read better with the values inverted, so that they are in ascending order.

Line 112 – it may sound better to remove “can be measured”, such that the phrasing is less redundant.

Line 118 – “higher specificity to…” may read better as “higher specificity than…”

Line 124 – “low-resource settings,” should be “low-resource settings;”

Line 139 – “in PLHIV in…” may read better as “for PLHIV in…”

Table 1 – watch for spaces before colons throughout main results column; spacing throughout makes the table hard to read.

Lines 174-175 – sentence with reference [43] is unclear. Out of more than 30 million eligible patients, fewer than 9 million PLHIV received TPT between 2018 and 2020?

Line 185 and Line 189 – “PLVIH” should be “PLHIV”

Line 209 – add comma after ART

Line 215 – remove space before comma after ART

Line 242 – citation formatting inconsistent with rest of paper

Line 255 – “being” may read better as “to be”

References – WHO Global Tuberculosis Report seems to be referenced thrice in different ways in references 1, 7, and 44. Are these actually different sources?

Author Response

Minor:

Line 32 – Perhaps specify latent infection vs active infection?

This was added to line 32 to clarify.

Lines 37-38 – “in sub-Saharan Africa” is redundant given context of sentence.

We have deleted “in sub-Saharan Africa”.

Line 40 – Sentence missing subject. Who is 19 more likely to develop TB?

We have rephrased as follows: “PLHIV are 19 times more likely to develop TB disease than their seronegative counter-parts”

Line 50 – “options for TPT” may read better as “options available for TPT”

We have added the word “available”

Lines 87-88 and Lines 115-16 – Formatting of sensitivity and specificity inconsistent with rest of review. Is this how it was reported in the reference?

We have harmonized the formatting to show percentages

Line 91 – “and a negative predictive value” may read better as “and had a negative…”

We added “had” in the text

Line 96 – Given context of sentence, “94.3% versus 90.4%” may read better with the values inverted, so that they are in ascending order.

We inverted the order of the percentages

Line 112 – it may sound better to remove “can be measured”, such that the phrasing is less redundant.

We removed “can be measured”

Line 118 – “higher specificity to…” may read better as “higher specificity than…”

We replaced “to” with “than”

Line 124 – “low-resource settings,” should be “low-resource settings;”

We changed “,” with “;”

Line 139 – “in PLHIV in…” may read better as “for PLHIV in…”

We replaced “in” with “for”

Table 1 – watch for spaces before colons throughout main results column; spacing throughout makes the table hard to read.

We will work with the journal to better format Table 1

Lines 174-175 – sentence with reference [43] is unclear. Out of more than 30 million eligible patients, fewer than 9 million PLHIV received TPT between 2018 and 2020?

This is indeed the meaning of the sentence, we have replaced “over” with “out of”

Line 185 and Line 189 – “PLVIH” should be “PLHIV”

We have corrected the typo

Line 209 – add comma after ART

We added a comma

Line 215 – remove space before comma after ART

We removed the space

Line 242 – citation formatting inconsistent with rest of paper

We have adapted the citation formatting

Line 255 – “being” may read better as “to be”

We replaced “being” with “to be”

References – WHO Global Tuberculosis Report seems to be referenced thrice in different ways in references 1, 7, and 44. Are these actually different sources?

We have corrected the references